# Stable Chinese Hamster Ovary Suspension Cell Lines Harboring Recombinant Human Cytochrome P450 Oxidoreductase and Human Cytochrome P450 Monooxygenases as Platform for In Vitro Biotransformation Studies

**DOI:** 10.3390/cells12172140

**Published:** 2023-08-24

**Authors:** Christian Schulz, Natalie Herzog, Stefan Kubick, Friedrich Jung, Jan-Heiner Küpper

**Affiliations:** 1Fraunhofer Project Group PZ-Syn, Fraunhofer Institute for Cell Therapy and Immunology, Branch Bioanalytics and Bioprocesses (IZI-BB) Located at the Institute of Biotechnology, Brandenburg University of Technology Cottbus-Senftenberg, Senftenberg, Germany; jan-heiner.kuepper@b-tu.de; 2Institute of Biotechnology, Brandenburg University of Technology Cottbus-Senftenberg, Senftenberg, Germany; natalie.herzog@b-tu.de (N.H.); friedrich.jung@b-tu.de (F.J.); 3Fraunhofer Institute for Cell Therapy and Immunology (IZI), Branch Bioanalytics and Bioprocesses (IZI-BB), Potsdam, Germany; stefan.kubick@fu-berlin.de; 4Institute of Chemistry and Biochemistry, Freie Universität Berlin, Berlin, Germany; 5Faculty of Health Sciences, Joint Faculty of the Brandenburg University of Technology Cottbus-Senftenberg, The Brandenburg Medical School Theodor Fontane and the University of Potsdam, Potsdam, Germany

**Keywords:** Chinese hamster ovary cells, CHO-K1, CPR, CYP450, cytochrome P450 monooxygenase, liver, NADPH P450 oxidoreductase, phase-1 biotransformation, serum-free, suspension cells

## Abstract

In the liver, phase-1 biotransformation of drugs and other xenobiotics is largely facilitated by enzyme complexes consisting of cytochrome P450 oxidoreductase (CPR) and cytochrome P450 monooxygenases (CYPs). Generated from human liver-derived cell lines, recombinant in vitro cell systems with overexpression of defined phase-1 enzymes are widely used for pharmacological and toxicological drug assessment and laboratory-scale production of drug-specific reference metabolites. Most, if not all, of these cell lines, however, display some background activity of several CYPs, making it difficult to attribute effects to defined CYPs. The aim of this study was to generate cell lines with stable overexpression of human phase-1 enzymes based on Chinese hamster ovary (CHO) suspension cells. Cells were sequentially modified with cDNAs for human CPR in combination with CYP1A2, CYP2B6, or CYP3A4, using lentiviral gene transfer. In parallel, CYP-overexpressing cell lines without recombinant CPR were generated. Successful recombinant expression was demonstrated by mRNA and protein analyses. Using prototypical CYP-substrates, generated cell lines proved to display specific enzyme activities of each overexpressed CYP while we did not find any endogenous activity of those CYPs in parental CHO cells. Interestingly, cell lines revealed some evidence that the dependence of CYP activity on CPR could vary between CYPs. This needs to be confirmed in further studies. Recombinant expression of CPR was also shown to enhance CYP3A4-independent metabolisation of testosterone to androstenedione in CHO cells. We propose the novel serum-free CHO suspension cell lines with enhanced CPR and/or defined CYP activity as a promising “humanised” in vitro model to study the specific effects of those human CYPs. This could be relevant for toxicology and/or pharmacology studies in the pharmaceutical industry or medicine.

## 1. Introduction

The liver represents the most important human organ for the biotransformation of drugs and xenobiotics due to the high phase-1 and -2 enzyme expression in hepatocytes [1]. Phase-1 reactions are the initial step in the metabolisation and elimination of drugs. Localised in the endoplasmic reticulum of hepatocytes, these reactions are facilitated by the complexation of cytochrome P450 oxidoreductase (CPR) and cytochrome P450 monooxygenases (CYPs) [2,3]. The latter make up a highly diverse superfamily with CYPs from families 1 to 3 being responsible for the oxidative metabolisation and clearance of a major part of clinically applied drugs [4,5,6,7]. The microsomal flavoprotein CPR is essential as a CYP electron donor in phase-1 metabolism [8,9,10]. In this NADPH-dependent oxidative process, CYPs mediate the sequential transfer of activated oxygen and electrons to a substrate [11,12]. For the development of new drug candidates, knowledge about their pharmacokinetics, toxicology, as well as their liver metabolites, are of great importance. Preclinical drug screening requires reliable in vitro cell and/or microsomal systems characterized, e.g., by highly specific phase-1 biotransformation properties to identify main and possible side metabolisation pathways at an early stage [4,5,13]. Of interest is also the in vitro generation of therapeutically active drug metabolites that could substitute prodrug administration and thus bypass the liver first-pass effect. 

Mammalian cell lines from, e.g., human embryonic kidney (HEK) or Chinese hamster ovary (CHO) are frequently used for the production of human proteins including their posttranslational modifications [14,15,16,17,18]. Those mammalian cell lines are primarily cultivated in serum-free suspension culture for recombinant protein expression, typically used in biotechnological production processes. One of the outstanding advantages is high cell densities typically achieved in stirred tank bioreactors and wave bioreactors resulting in high-yield protein production. The absence of animal sera enables fast and convenient isolation of secreted proteins required for therapeutic use [19]. CHO cells are genetically stable and highly proliferative, allowing for easy handling and modification in scale-up processes. A major advantage of CHO cells is that they are recognised as safe with respect to infectious and pathogenic agents and are, thus, well accepted by regulatory authorities. 

With regard to the recombinant mono- and co-expression of phase-1 enzymes, such as CPR and CYPs, a variety of in vitro models have been developed for the investigation of liver-specific biotransformation as well as for toxicological or metabolisation studies. Recent examples include studies on the effects of CYP1A1*13 polymorphism on holoenzyme expression and activity or effects of CPR/CYP1B1 co-expression on the metabolisation of 8-methoxypsoralen in *E. coli* expression systems in the context of photochemotherapy [20,21]. Co-expression of CPR, CYPs, and cytochrome b5 was performed in HEK293 cells to investigate the regulatory properties of CPR on the expression and activity of certain CYPs [14,22]. Cytochrome b5 is, along with CYPs, a physiological redox partner of CPR and can, thus, exert modulating effects on the activity of CYPs. Several hepatic carcinoma cell line-based systems, such as HepG2 or HepaRG, have been generated in recent years, overexpressing various CYPs as well as CPR/CYPs and showing high phase-1 activities [23,24,25,26,27,28,29]. Limitations of these models stem from the hepatic origin of the cells and comprise background activities of nonrecombinant CYPs depending on the human source and culture conditions [30,31]. An interesting scientific approach is given by the therapeutic applications of mesenchymal stem cells modified by lentiviral gene transfer with CPR and CYPs for gene-directed enzyme prodrug therapy (GDEPT) [32]. A promising model for specific recombinant expression of phase-1 enzymes is Chinese hamster cell lines, such as V79 hamster lung fibroblasts or CHO cells. These have long been established for the cell-based and cell-free recombinant expression of complex human proteins [21,33,34,35]. It has been shown that CHO cells are well suited for recombinant expression of human phase-1 and -2 liver enzymes with no basal CYP background activity described so far [21,33,34,36]. Potential limitations are low basal CPR and cytochrome b5 expression and activity compared with hepatocytes [33,37]. This might have a modulating and/or limiting effect on CYP holoprotein content and activity in a CYP-dependent manner [14,33,36]. Since the 1990s, several studies with genetically modified V79 cells have been developed on liver enzyme-dependent drug conversion and CPR/CYP polymorphism [38,39,40,41]. Based on the work of Ding et al., CHO-DUKXB11 cells have been used for co-expression of CPR and CYPs [33,36]. These cell lines generated by Ca^2+^-phosphate transfection showed significant CPR and CYP activity and have recently been used for metabolisation studies [21,34]. However, a limitation of those models is the serum-dependent cultivation in monolayer culture, which complicates upscaling and use of generated metabolites for therapeutic applications due to supplements of animal origin. 

In this study, we pursued the approach of lentivirus-mediated gene transfer of phase-1 enzymes into CHO-K1 cells (hereinafter referred to as CHO). The aim was to achieve stable and specific high-yield phase-1 enzyme expression and activity on the single cell level, by using CMV promotor-controlled cDNAs of human CPR in combination with CYP1A2, CYP2B6, or CYP3A4 on the background of highly proliferative, serum-free growing CHO suspension cells. The recombinant expression systems described here should enable the production of functional phase-1 enzymes as well as metabolites of drugs for toxicological studies and industrial/pharmaceutical applications.

## 2. Materials and Methods

### 2.1. Cell Culture and Generation of Recombinant CYP or CPR/CYP Expressing CHO Cells

Suspension-adapted Chinese hamster ovary cells (CHO-K1 suspension cells kindly provided by the Fraunhofer IZI-BB, Potsdam, Germany) were routinely cultivated in ProCHO5 medium (Lonza Group AG, Basel, Switzerland) supplemented with 6 mM L-alanyl-L-glutamine (Biowest, Nuaillé, France). CHO suspension was cultured in 125 mL baffled flasks (30 mL culture, Corning, NY, USA) at 37 °C and 5% CO_2_ at 120 rpm on an orbital shaker. For small-volume suspension culture (e.g., during cloning), cultivation was carried out in polystyrene low attachment plates (6-, 24-, 96-well, Corning). For adherent culture, required for the selection of single clones, 2% FBS (Bio&Sell GmbH, Feucht, Germany) was added to the medium, and cultivation was carried out in TCP vessels for adherent cell culture (STARLAB GmbH, Hamburg, Germany).

Invitrogen™ Gateway^®^ Technology (Thermo Fisher Scientific Inc., Waltham, MA, USA) was used for constructing expression plasmids of CYP and CPR/CYP. Initially, an entry vector (GeneCopoeia™, Rockville, MD, USA) with the coding sequence for human CPR (NCBI reference sequence: NM_000941.2) delimited from attachment sites (attL) was applied to recombine the CPR cDNA into the attR-containing lentiviral expression vector pLenti6/V5-DEST (Thermo Fisher Scientific Inc.) using LR recombination reaction. Accordingly, CPR cDNA expression was regulated by the human cytomegalovirus (CMV) promoter. The vector also contained a Blasticidin resistance gene for the selection of successfully transfected clones. Standard methods, including the helper cell line 293FT, were applied for generating recombinant lentiviruses with the ViraPower™ Lentiviral Expression System (Thermo Fisher Scientific Inc.; R70007). The cell line 293FT was cultured in Dulbecco’s MEM supplemented with 10% FBS, 4 mM L-alanyl-L-glutamine, 0.1 mM nonessential amino acids, 1 mM sodium pyruvate, and 500 μg/mL geneticin^®^ (all from Bio&Sell GmbH) at 37 °C and 5% CO_2_. Co-transfection of the vector plasmid pLenti6/V5-DEST-CPR together with helper plasmids pLP1, pLP2, pLPVS/VG occurred via Lipofectamine™ 2000 (all Thermo Fisher Scientific Inc.) into 293FT. Twenty-four hours after transfection, the culture medium was replaced by a fresh medium. After 48 h incubation, recombinant lentivirus was collected from the supernatant, concentrated by centrifugation with Vivaspin^®^ 20 columns (Merck KGaA, Darmstadt, Germany), and used to infect CHO parental cells. Briefly, CHO cells in suspension were transferred with a density of 1 × 10^5^ cells/well in nonadherent 24-well plates (Corning). Then, the lentivirus suspension and hexadimethrine bromide (6 μg/mL, Merck KGaA) were added. After 24 h, the medium was replaced with a fresh culture medium. Forty-eight hours postinfection, selection for successfully engineered CHO-CPR cells was initiated by adding 3 µg/mL Blasticidin (AppliChem GmbH, Darmstadt, Germany). Within two weeks, Blasticidin-resistant cells were expanded. For single clone isolation, decadal dilution of CHO-CPR mix clones (from 5 × 10^5^ till 5 × 10^2^ cells/well) was seeded in 6-well plates, and grown isolated colonies were picked and expanded. The growth of suitably isolated colonies (1–2 mm^2^) required about 2 weeks. After picking colonies using cloning discs (Ø~2 mm, Merck KGaA) soaked in trypsin-EDTA solution and transferring them to 96-well plates, the cells were gradually expanded to T150 format under selection conditions. Both selection and single clone isolation were performed in adherent culture. Subsequently, the cell clones were transferred into suspension by using a serum-free culture medium and cultivation in baffled flasks. Finally, CHO-CPR single clones were transferred into suspension and screened for CPR expression/activity using the methods described below. The generation of CYP-overexpressing CHO clones with and without recombinant CPR expression followed the same procedure. The cDNA of three prominent human CYP enzymes (CYP1A2-NCBI ref. seq.: BC067428.1; CYP2B6-NCBI ref. seq.: BC067430.1; CYP3A4-NCBI ref. seq.: NM_017460.6) was cloned into the lentiviral expression vector pLenti4/V5-DEST (Thermo Fisher Scientific Inc.) with a Zeocin resistance gene, respectively. Recombinant lentiviruses were used to infect CHO parental cells and the best-performing CHO-CPR C12 clone. Selection of CHO-CYP, as well as CHO-CPR/CYP clones, was performed by culturing with 300 µg/mL Zeocin (Life Technologies GmbH, Darmstadt, Germany) or Blasticidin/Zeocin double selection.

### 2.2. Population Doubling Time

For determination of the population doubling time (PDT), maximal cell density, and vitality over time, CHO cell clones were seeded in suspension with a density of 0.5 × 10^6^ cells/mL in a 125 mL baffled flask (30 mL culture volume) and cultured under standard conditions with respective selection antibiotics. On a daily basis, the cells were counted in duplicate with a Countess^®^ II FL automated cell counter (Thermo Fisher Scientific Inc.) and Trypan blue staining to distinguish between intact and damaged cells as a measure to assess cell viability. Population doubling period was determined using the formula PDT = (t_2_ − t_1_) × log2/log(q_2_/q_1_), with t_1_ − t_2_ = period of steady growth rate in hours, q_1_ = vital cell number at t_1_, and q_2_ = vital cell number at t_2_.

### 2.3. Gene Expression Analysis by Quantitative Real-Time PCR

Total RNA was extracted from cell pellets (2 × 10^6^ cells) using the innuPREP RNA Mini Kit (Analytik Jena AG, Jena, Germany) according to the manufacturer’s protocol. RNA purity and amount were determined by spectrophotometry with a NanoDrop™ 1000 (Thermo Fisher Scientific Inc.). To remove traces of genomic DNA, the DNA-free kit was used for DNase digestion. (Ambion, Life Technologies GmbH). Subsequently, the integrity of isolated RNA (28S and 18S rRNA bands) was verified by agarose gel electrophoresis and ethidium bromide staining. A total of 2 µg isolated mRNA was used for reverse transcription with the RevertAid H Minus Reverse Transcriptase (Thermo Fisher Scientific Inc.), supplemented with dNTPs (1 mM, Carl Roth GmbH + Co. KG, Karlsruhe, Germany) and oligo(dT)_18_ primers (25 ng/µL, Thermo Fisher Scientific Inc.). In addition, to exclude contamination with genomic DNA, a reverse transcriptase minus control (-RT) was created. Finally, + and -RT cDNA were 1:10 diluted in DEPC water (Carl Roth GmbH + Co. KG) and stored at −80 °C until use. Quantitative real-time PCR (qRT-PCR) was performed with Maxima Probe qPCR Master Mix (Thermo Fisher Scientific Inc.) and intercalating dye Evagreen (Biotium Inc., Fremont, CA, USA). Primers for target genes CPR, CYP1A2, CYP2B6, and CYP3A4 were purchased from BioTeZ Berlin Buch GmbH (Berlin, Germany), and CHO reference genes GAPDH-CHO and Vezt [42] obtained from Thermo Fisher Scientific Inc. were used in a final concentration of 0.25 µM. Corresponding primer sequences are shown in Table 1.

Gene expression analysis was realised using a CFX96 real-time system (Bio-Rad Laboratories, Inc., Hercules, CA, USA). Additional determination of GAPDH-CHO and Vezt expression allowed normalisation by 2^(−ΔΔCt)^ method. Except for -RT and NTC (no template control), sample gene expression levels were determined in duplicates in three independent qPCR runs (n = 6). Verification of gene-specific amplicons was proved by performing a melting curve analysis after each qRT-PCR experiment.

### 2.4. Protein Expression Analysis by Immunodetection

Proteins for immunodetection by Western blot analysis were isolated from cell pellets (2 × 10^6^ cells) by incubation in RIPA buffer (including PMSF, 1 mM) on ice (Carl Roth GmbH + Co. KG) for 15 min, followed by centrifugation for 10 min at 10,000 rpm and 4 °C. The Pierce™ BCA Protein Assay Kit (Thermo Fisher Scientific Inc.) was used for protein quantification, followed by 1:5 dilution in 6× Laemmli buffer, denaturation for 5 min at 95 °C, and storage at −20 °C until use. SDS-PAGE was conducted following the protocols established by Laemmli et al. [43]. Briefly, 20 µg proteins were loaded per lane onto a 10% resolving gel. A 1:1 mixture of MagicMark™ XP Western Protein Standard and Page Ruler™ Plus prestained was used for molecular weight allocation (both Thermo Fisher Scientific Inc.). After blotting on a PVDF membrane (Carl Roth GmbH + Co. KG), the blot was cut into several parts to allow parallel detection of several proteins from one blot. The visible marker bands of the Page Ruler™ Plus prestained were used as a reference. For parallel detection of CPR (~80 kDa), CYP (~55 kDa), and the loading control GAPDH (~35 kDa), the blot was divided slightly below the 70 kDa and centrally between the 55 and 35 kDa marker bands. Immunodetection was then performed separately for each blot part with the corresponding antibodies. Initially, a 2% BSA solution (solved in PBS, Carl Roth GmbH + Co. KG) was used for blocking of unspecific binding sites at ambient temperature for 1 h. Immunodetection of CPR was realised by using a polyclonal rabbit-anti-CPR IgG primary antibody (1 mg/mL, Abcam, Cambridge, UK) diluted 1:1000 in 2% BSA. CYP detection was realised by using monoclonal mouse primary antibodies diluted 1:1000 in 2% BSA [anti-CYP1A2 IgG1 (clone: 3B8C1, 0.2 mg/mL, Thermo Fisher Scientific Inc.); anti-CYP2B6 IgG2a (clone: 3D5, 1 mg/mL, Biotechne, Minneapolis, MN, USA); and anti-CYP3A4 IgG1 (clone: 3H8, 1 mg/mL, Biomol, Hamburg, Germany)]. Primary antibody of the loading control GAPDH was monoclonal mouse-anti-GAPDH IgG (0.5 mg/mL, antibodies-online GmbH, Aachen, Germany) in 1:5000 dilution. Primary antibody binding was performed at 4 °C overnight. Protein assessment by enhanced chemiluminescence reaction was facilitated by incubating the blots with peroxidase-conjugated secondary antibody goat-anti-rabbit IgG or goat-anti-mouse IgG (both 1:2000 in 2% BSA in PBS, Merck KGaA) at ambient temperature for 1 h and detection by using the Amersham Prime Western blotting detection reagent (GE Healthcare, Chicago, IL, USA) combined with a Biostep Celvin^®^ S 420 chemiluminescence imaging system (Biostep GmbH, Burkhardtsdorf, Germany).

For indirect immune fluorescence staining, adherent CHO cells growing on glass slides (Thermo Fisher Scientific Inc.) were fixed in a 5% formaldehyde solution for 30 min. Permeabilisation was achieved in 0.2% Triton-X100 solution for 10 min, followed by blocking of unspecific binding sites with a 3% BSA solution (solved in PBS) for 30 min. All work steps were carried out at room temperature. CPR and CYP protein binding was visualised using the same primary antibodies mentioned for immunodetection at 1:100 dilution. Secondary antibodies for fluorescence detection were a Cy3-conjugated goat-anti-rabbit IgG (1:200, Dianova GmbH, Hamburg, Germany) for CPR and a FITC-conjugated polyclonal donkey-anti-mouse IgG (H+L) FITC (1:200, Life Technologies GmbH) for CYPs. The binding of the primary antibody was performed overnight at 4 °C. For secondary antibody binding, samples were incubated at ambient temperature for 1 h. In between the antibody binding steps, samples were washed 4 × 15 min with PBS. Potential nonspecific background binding of the antibodies was monitored by appropriate controls. These were either stained with an isotypical primary antibody (for CPR: polyclonal rabbit IgG, 1 mg/mL, Abcam) without target-binding properties or, for CYP staining, by omitting the primary antibody. Genomic DNA was stained by 5 min incubation with 4′,6-diamidino-2-phenylindole dihydrochloride (DAPI, 0.2 µg/mL in PBS, Carl Roth GmbH + Co. KG). Microscopic visualisation was realised using an LSM800 confocal laser scanning Microscope and ZEN 2.3 software for picture postprocessing (Carl Zeiss Microscopy GmbH, Jena, Germany).

### 2.5. Enzyme Activity Determination of CPR and CYP450

For CPR activity determination, CHO microsomal fractions (MF) were isolated from 1 × 10^7^ cells. Cell lysis was performed by sonication using a Sonopuls ultrasonic homogeniser equipped with a UW3100 ultrasonic probe (BANDELIN electronic GmbH & Co. KG, Berlin, Germany) in an ice bath. Cells were treated with 3 × 10 s pulses at 40% amplitude. To prevent overheating of the proteins within the samples, pulses were intermitted by 20 s breaks. Microsomal extraction was subsequently conducted with the microsome isolation kit (BioVision Inc., Milpitas, CA, USA) according to the manufacturer’s recommendations. MF were eluted in 200 µL cold storage buffer, aliquoted, and stored at −80 °C until use. One aliquot was used for the determination of protein content by quantification against a BSA standard with the Pierce™ BCA Protein Assay Kit. As a positive control for CPR activity, MF from HepG2 cells were isolated in parallel. For analysis of CPR activity in MF, the cytochrome P450 reductase activity kit (Abnova Corporation, Taipei City, Taiwan) was used. Additionally, the kit included a positive control (recombinant human CPR; rhCPR) and HepG2-MF; commercially available primary human hepatocyte microsomes (HHM, Thermo Fisher Scientific Inc.) served as a reference. Prior to CPR activity studies with MF, standards with the substrate glucose-6-phosphate (G6P) were separately measured for later MF-based CPR activity calculation. Microsomal CPR activity in CHO and CHO-CPR MF was determined in duplicates, whereby for each sample 5 µg protein were used. Data were collected with n = 11 from three independent experiments. To differentiate between CPR and the background activity of residual substrate-reducing enzymes, MF were supplemented with the CPR inhibitor diphenyleneiodonium chloride (DPI, 0.1 µM). CPR-mediated reduction of the substrate was detected at OD_460nm_ with a FLUOstar Omega microplate reader (Software version: 3.00 R2, BMG LABTECH GmbH, Ortenberg, German) for 30 min. For subsequent data analysis, the MARS data analysis software (Version: 2.41) was used. 

For the assessment of CYP activity in CHO cells, the cells were analysed with P450-Glo™ CYP induction/inhibition assays (Promega, Madison, WI, USA). Data were collected with n = 12 from two independent experiments. Briefly, adherent CHO cells with specific expression of either CYP or CPR/CYP were incubated with 50 μL of the respective luminogenic CYP substrate Luciferin-1A2, Luciferin-2B6, or Luciferin-IPA for CYP3A4 diluted in PBS at 37 °C, 5% CO_2_ for 60 min. Hereafter, 25 μL of the supernatant was pipetted into a white-walled 96-well plate (SARSTEDT AG & Co. KG, Nümbrecht, Germany), mixed with 25 µL of luciferin detection reagent and incubated at ambient temperature for 20 min. Luminescence was detected with a FLUOstar Omega microplate reader (BMG LABTECH GmbH). Data analysis was performed using the MARS data analysis software (Version: 2.41). Additionally, the remaining cells and substrate solution in the original 96-well plate were mixed with 25 μL ATP reagent of the CellTiter-Glo^®^ 2.0 assay (Promega) and stored in the dark for 10 min. ATP levels were detected by measuring luminescence to allow for the normalisation of the effective cell number. For each specific CYP detection, a corresponding HepG2-CYP overexpressing cell line was included as a positive control, as well as parental CHO and CHO-CPR C12 cells as references.

### 2.6. Metabolisation Studies with Prototypical CYP-Substrates

In preliminary studies concerning the readaptation of the CHO-CPR/CYP models to suspension as well as parallel experiments with isolated microsomes, different supplementation studies were performed to increase CYP activity (unpublished data). Based on this, a pretreatment protocol for CHO cells was established, which was also used for the metabolisation studies with prototypical CYP substrates shown here. It includes the pretreatment with 2% DMSO in the culture medium for 24 h prior to metabolisation. Pretreatment and metabolisation of prototypical CYP substrates were performed in a 10 mL suspension culture (2 × 10^6^ cells/mL) with CHO-CPR/CYP1A2 C9 and CHO-CPR/CYP3A4 C1. During both experimental sections, cell suspensions were incubated under culture conditions and orbital shaking at 150 rpm in 50 mL Duran laboratory bottles (VWR, Radnor, PA, USA). The specific CYP activity of the CHO-CPR/CYP cell lines was investigated comparatively with parental cell lines CHO and CHO-CPR C12. Metabolisation of testosterone for the CPR/CYP3A4 and phenacetin for the CPR/CYP1A2 clone (both 100 µg/mL, Merck KGaA, Darmstadt, Germany) was performed after DMSO pretreatment for up to 48 h. Metabolites formed were quantitatively detected by HPLC at different time points (0, 4, 24, 48 h) and cell viability was monitored in parallel. CYP activity of the investigated cell lines was calculated from whole protein content after 48 h metabolisation, using Pierce™ BCA Protein Assay Kit (see Section 2.4).

Metabolite profiling and quantification were based on HPLC analysis of culture supernatants. After collection, supernatants were mixed 1:1 with ice-cold acetonitrile (VWR, Radnor, PA, USA), vortexed, incubated on ice for 5 min, and centrifuged at 16,000× *g* for 15 min. Supernatants obtained were analysed by HPLC. The setup consisted of an SCL-10AvP (SHIMADZU, Kyoto, Japan) with a ZORBAX SB-C18 column (Agilent Technologies, Santa Clara, CA, USA). For metabolite separation, a 20 µL sample was separated in a gradient of KH_2_PO_4_ buffer (10 mM, pH 3, mobile phase A) and acetonitrile (mobile phase B) at a flow rate of 1 mL/min. Metabolites were detected based on elution time using DAD (SPD-M10AvP) at 245 nm. The gradients for testosterone (CYP1A2) were as follows: 0–5 min 5% B, 15 min 40% B, 16 min 90% B, 20 min 90% B, 24 min 5% B, 25 min 5% B; for phenacetin (CYP3A4): 0–3 min 30% B, 20.5 min 55% B, 22.5 min 55% B, 23 min 95% B, 26 min 95% B, 26.5 min 20% B, 29 min 20% B, 30 min 30% B. Quantification of detected metabolites was performed using standard solutions of commercially available reference substances (6β-hydroxytestosterone (6β-OH-T); 4-androstene-3,17-dione; acetaminophen all from Merck) in the range of 1–50 µM. Stock solutions of substrates and references were dissolved in methanol (Honeywell International Inc., Charlotte, NC, USA).

### 2.7. Statistical Analysis

Unless otherwise described, one-way ANOVA with Tukey’s multiple comparison test was used to determine significant differences between groups. Prism 6 (GraphPad Software, San Diego, CA, USA) was used for the calculation. For all statistical analyses statistical significance was assumed for *p* < 0.05.

## 3. Results

### 3.1. Recombinant CPR Expression and Activity in Selected CHO-CPR Clones

Upon lentivirus CPR cDNA transduction, 35 individual clones were isolated, and CPR expression was initially characterised at the protein level by immunodetection. For CHO-CPR clones, two bands for CPR were visible between 75–80 kDa. The weak lower band corresponded to basally expressed CPR, being also detectable in parental CHO cells. The other band at slightly increased molecular weight varied in intensity between the clones and corresponded to recombinantly expressed human CPR, which additionally carried a V5 tag. Clones 10, 12, and 13 were found to display the highest recombinant CPR protein expression and were further characterised (Figure 1B). Studies on CPR mRNA expression showed a significantly increased expression in clones 10, 12, and 13 compared with parental CHO cells (Figure 1A). The level of CPR mRNA expression varied between the selected clones, with CHO-CPR C12 showing the highest expression (CHO_CPR C10_ = 857 ± 285 fold, CHO_CPR C12_ = 3574 ± 906 fold, CHO_CPR C13_ = 2199 ± 370 fold; normalised to basal CPR expression in CHO cells). Indirect immunofluorescence staining confirmed the increased CPR expression compared with parental CHO cells, with almost all visible cells showing an increased CPR signal (Figure 1D). Morphologically, minor changes were observed for the selected clones, both with fluorescence and light microscopy in phase contrast mode. Morphologically, the most striking feature of clone 13 was the appearance of slightly enlarged cells in adherence and suspension as well as the increased occurrence of multinucleated cells. Analyses of proliferation capacity in suspension showed differences in the population doubling time and maximum cell density between the selected clones (Table 2). With Blasticidin selection, CPR clones C10 and C13 showed a clearly prolonged doubling time and lower maximum cell density compared with CHO cells (without selection). CHO-CPR C12 with 19.9 h was the closest to CHO cells (18.3 h). Likewise, this clone reached the highest cell density with 5.9 ± 0.3 × 10^6^ cells/mL after 120 h cultivation.

As a decision criterion for the selection of a CHO-CPR clone prior to further modification with human CYP enzymes, the functionality of recombinantly expressed human CPR was quantified by enzyme activity assays from isolated microsomal fractions (Figure 1C). All selected CPR clones showed significantly increased CPR activity compared with CHO cells. Clone 12 showed an ~8-fold increase in CPR activity (CHO_CPR C12_ = 81.3 ± 34.0 mU/mg_protein_ vs. CHO_basal_ = 10.9 ± 6.4 mU/mg_protein_; *p* < 0.0001), which was the highest activity of all clones. However, the CPR activity was always significantly lower than in HepG2 (119.5 ± 17.1 mU/mg_protein_) and HHM (149.4 ± 7.8 mU/mg_protein_).

### 3.2. Specific Recombinant CYP Expression in CHO Cells with or without Recombinant CPR

CHO parental cells were transduced by recombinant lentiviruses to express human CYP1A2, -2B6, and -3A4, respectively. Furthermore, supertransfectants with the above-mentioned CYPs based on CHO-CPR C12 were generated. To identify suitable individual clones, the recombinant CYP protein expression of the corresponding CHO-CYP and CHO-CPR/CYP variants were analysed by Western blotting. Parental CHO, CHO-CPR C12 as well as previously generated HepG2 cells with recombinant CYP overexpression were used as controls. From the variety of isolated CHO-CYP1A2, -2B6, and -3A4 clones as well as corresponding CHO-CPR/CYP variants, two clones of each were selected according to their overexpression (Figure 2B). In parental CHO cells, a weak protein band was detected exclusively for CYP2B6. Gene expression studies confirmed the observations made by protein immunodetection (Figure 2A). The CPR/CYP supertransfectants showed CPR expression levels higher than in CHO cells and as high as in their CHO-CPR C12 parental cells. With regard to CYP expression, no substantial differences were determined at the mRNA level between CYP transfectants and CPR/CYP supertransfectants. All clones showed a distinct mRNA expression of the respective transfected CYP enzyme.

Recombinant CPR and CYP expressions were further investigated by indirect immunofluorescence (Figure 3). Results obtained by Western blotting were confirmed, whereby recombinant CPR protein expression was increased for all CHO-CPR/CYP clones. CHO-CYP clones showed basal CPR expression. In modified CHO and CHO-CPR cells, specific expression of CYP1A2, -2B6 as well as -3A4 was clearly visible in most cells. CYP signals appeared most intense near the nucleus and co-localised with CPR. Differences in CYP signal intensities were less pronounced between clones of the same modification using this method. Only CHO-CYP1A2 showed a difference with a higher signal strength for the C2 clone.

### 3.3. Proliferation Capacity of CHO-CYP and CHO-CPR/CYP Clones in Suspension

Examination of the proliferative capacity in the suspension of selected CHO clones revealed differences between the generated cell lines (Table 3). It should be noted that all clones showed a high vitality of ≥90% over a cultivation period of at least five days. The population doubling times varied in the range between 20 to 24 h and did not reveal any dependency on the expressed CYP. Similarly, the maximum cell densities achieved under the selected culture conditions varied between the CHO clones in the range between 3.2–6.9 × 10^6^ cells/mL. Except for CHO-CYP2B6 C1, CYP transfectants had a clearly prolonged proliferation doubling time and a lower cell density than parental CHO cells. The same can be seen in the supertransfectants with CPR/CYP co-expression regarding their CHO-CPR C12 parental cell line.

### 3.4. Specific CYP Activity in CHO-CYP Clones with and without Recombinant CPR

The selected CHO clones were examined with regard to their specific CYP activity (Figure 4). In all CHO clones modified with CYP1A2, this enzyme activity was clearly detectable. Interestingly, CYP1A2 enzyme activity in clone CHO-CYP1A2 C2 was at the same level as in the corresponding human CPR-overexpressing clones CHO-CPR/CYP1A2 C8 and C9. Clone CHO-CYP1A2 C4 showed lower CYP1A2 activity. In contrast, a comparison of CHO-CYP and CHO-CPR/CYP clones showed that CYP2B6 and -3A4 activity can be increased by simultaneous CPR overexpression. For the three CYPs investigated, there was no background enzyme activity detectable in CHO parental cells nor in the CHO-CPR clone 12.

### 3.5. Metabolisation of Prototypical CYP-Substrates in Suspension Culture

To prove the CYP-specific activity of generated CHO-CPR/CYP clones, metabolisation studies with prototypical CYP substrates were performed in suspension with CHO-CPR/CYP3A4 C1 and CHO-CPR/CYP1A2 C9 in comparison with parental CHO and CHO-CPR C12 cells. In these studies, the metabolisation of testosterone by the CPR/CYP3A4 and phenacetin by the CPR/CYP1A2 clone was monitored for up to 48 h. 

In the testosterone metabolisation study, two metabolites 6β-hydroxytestosterone (6β-OH-T; CYP3A4 main metabolite) and androstenedione were identified by HPLC from supernatants (Figure 5A). Notable amounts of 6β-OH-T were only detected for CHO-CPR/CYP3A4 C1 (57 ± 5 nmol after 48 h). The metabolisation kinetics showed an initially high rate of 6β-OH-T formation, which decreased steadily over time and stagnated visibly towards the end of metabolisation. Parental CHO and CHO-CPR C12 cells did not show considerable 6β-OH-T formation above the background. An interesting observation was the detection of androstenedione in all three cell lines. The formation rate as well as the effectively formed amount of androstenedione over time was the lowest in the CHO cells compared with recombinant human CPR-expressing ones. CHO-CPR C12 and CHO-CPR/CYP3A4 C1 showed more than 2-fold increase in androstenedione formations compared with CHO cells, whereby the CYP3A4 clone generated slightly more of this metabolite than CHO-CPR C12 (androstenedione after 48 h: n_BG_ = n.d.; n_CHO_ = 77 ± 10 nmol; n_CHO-CPR C12_ = 171 ± 9 nmol; n_CHO-CPR/CYP3A4 C1_ = 199 ± 12 nmol; BG: background with substrate without cells). Interestingly, with CHO-CPR/CYP3A4 C1, about 4 times more testosterone was metabolised to androstenedione than to the CYP3A4 metabolite 6β-OH-T. 

Likewise, phenacetin conversion by CHO-CPR/CYP1A2 C9, parental CHO, and CHO-CPR C12 cells was monitored for 48 h (Figure 5B). HPLC analysis of supernatants revealed that the main CYP1A2 metabolite was detected for CHO-CPR/CYP1A2 C9 only (668 ± 29 nmol after 48 h). Within the first 24 h, the conversion of phenacetin to acetaminophen was almost constant at a high level. In the period from 24 to 48 h, the formation rate decreased significantly.

## 4. Discussion

In the present study, CHO suspension cells were genetically modified by lentiviral gene transfer to generate CHO cell clones with specific phase-1 enzyme activity. The expression strategy involved sequential lentivirus transduction of human CPR cDNA followed by transduction of human CYP cDNAs for CYP1A2, CYP2B6, or CYP3A4, respectively. As controls, CHO cell lines with CYP but without additional CPR expression were generated. The aim was to create a panel of novel CHO cell lines with stable human CYP enzyme overexpression and with or without additional human CPR overexpression. Since the CHO parental cells did not show any measurable endogenous activities of the studied CYP enzymes, such a cell panel should represent a useful “humanised” in vitro model for background-free studies of the metabolisation activity of a specific CYP enzyme. This could be relevant for various toxicology and/or pharmacology studies in the pharmaceutical industry or medicine.

We performed a comprehensive analysis to select the best CPR clone, which then served as the basis for further modification with human CYPs. As a result, human CPR-overexpressing cell clones 10, 12, and 13 were subjected to more detailed characterisation. Here, almost all visible cells from the chosen CPR clones showed an intensive CPR immunofluorescence signal. As expected, the CPR activity in all CHO-CPR clones was significantly higher than in parental CHO cells, which showed some endogenous CPR activity. Compared with HHM, it should be noted that the measured CPR activity of CHO-CPR clones was strongly affected by the cell lysis procedure applied during microsome isolation. The cell disruption via ultrasound may have led to a reduction in the activity of microsomal enzymes. Thus, a comparison of our data with those of previous studies using human primary liver cells or HHM is limited, as information about the respective microsome isolation process was not disclosed in those studies [24,25,44].

An additional parameter important for the selection of the most promising CHO-CPR clone was the proliferation time. All generated cell clones showed a prolonged proliferation time and lower maximum cell density compared with parental CHO cells (Table 2). This might be due to the genetic modification itself, constitutive transgene expression, and/or Blasticidin selection [45,46]. Depending on the application, single cell performance in terms of, e.g., transgene expression and activity, as well as the entire cell culture performance (proliferation, max. cell density, vitality), might be important. All these were relevant factors for the selection of CHO-CPR C12 for further modification with human CYPs.

For the selection of the most effective CYP-expressing CHO-CPR/CYP or CHO-CYP clones, gene and protein expression levels were analysed. CYP mRNA expression in the CHO parental cells was at or below the detection limit (C_t_ > 50 cycles), indicating that there is almost no basal gene expression of the investigated CYPs. Regarding CPR, a weak and reproducible mRNA expression was always detectable in CHO cells confirming the results from protein analysis. In contrast, strong mRNA expression of corresponding CYPs was detected in all selected clones. Similar to CHO cells, CHO-CYP clones further showed some endogenous CPR mRNA expression, whereas it was strongly increased in CHO-CPR/CYP clones. For CYP enzymes, specific recombinant protein expression of the introduced genes was observed in all clones. Expression efficiency varied between individual clones as observed for CPR before. This was particularly evident for CYP3A4 expression of the selected CHO-CYP3A4 clones C1 and C4 and for CPR expression of the CHO-CPR/CYP3A4 clones C1 and C3. Interestingly, we found a weak CYP2B6 protein band in CHO parental cells. This was the only indication of basal CYP protein expression in CHO cells with respect to the CYPs studied. However, CYP2B6 enzyme activity could be detected in neither CHO cells nor the CHO-CPR cell clone. An interesting finding was that simultaneous recombinant expression of CPR and CYP in CHO cells seemed to positively influence the activity of some CYPs more than others. This difference was most pronounced in clones with recombinant CYP3A4 expression, followed by CYP2B6. In clones with recombinant CYP1A2 expression, an influence of the artificially increased CPR expression was only very slightly evident. One reason for this could be clonal variability, which cannot be excluded due to the lentiviral cloning procedure [45,47]. However, it is well conceivable that total CPR protein expression levels could be a limiting factor for some CYP-dependent oxidative substrate conversions [33]. An increase in available CPR results in a higher conversion rate since co-localisation with CYPs is more likely and electrons necessary for the reaction can be transferred more effectively via the electron transport chain. Thus, artificial CPR expression shifts the quantitative ratio between CPR and CYP in favour of faster substrate conversion. Apart from the amount of CPR available, it is also likely that CPR has a CYP-dependent regulatory influence on the formation of holoenzymes at the translational or protein level. Such regulatory processes have already been reported [14,36,48]. We need more information to better understand the observed differential dependence of CYP activities associated with recombinant CPR expression and to exclude CYP-independent clonal variations as the lone cause for this. 

Investigations of the selected CYP-expressing CHO variants consistently showed a slightly prolonged generation time and reduced maximum cell density with regard to their proliferation potential in suspension compared with the respective parental cell line (CHO or CHO-CPR C12). The cell densities achieved in suspension varied among individual clones between 3.2 × 10^6^ and 6.9 × 10^6^ cells/mL with cell viabilities of at least ≥ 90%. Thus, all selected CYP-expressing CHO cell lines can be cultivated effectively.

An optimal cell model to investigate the particular role of a recombinantly expressed CYP in drug metabolism would be based on a parental cell line that does not contain measurable enzyme activity of any CYP relevant to the drug(s) of interest. However, in addition to hepatic cells, many commonly used cell lines comprise activities of one or more drug-metabolising CYP enzymes as we previously have found (unpublished data) and that have been also described by others [49]. In order to gain information about this in the CHO model and to also investigate the performance at the level of intact suspension cells, conversion studies with prototypical substrates were performed with two CHO-CPR/CYP clones for up to 48 h. These studies were performed in comparison with parental CHO and CHO-CPR C12 cells in order to be able to differentiate between potential basal and recombinantly induced metabolisation. Comparing the conversion of phenacetin by CHO-CPR/CYP1A2 C9 with parental cells, acetaminophen formation was detectable only in the CYP1A2-overexpressing clone. No background metabolisation of phenacetin was observed in the parental cell lines. Using CHO-CPR/CYP1A2 C9 suspension cells, a yield of ~670 nmol of the main CYP1A2 metabolite acetaminophen was obtained after 48 h, whereby ~550 nmol was generated within the initial 24 h. Comparing the calculated metabolisation rates with human hepatocyte systems, currently used for in vitro toxicological analyses, the CHO-CPR/CYP1A2 C9 cells with 271 ± 50 pmol/mg_protein_·min after 4 h and 206 ± 28 pmol/mg_protein_·min after 24 h show a significantly higher CYP1A2 activity than, for example, HepaRG 17.5–154.6 pmol/mg_protein_·min and HepG2-CYP1A2 with 120 pmol/mg_protein_·min [50]. The CYP1A2 activity of freshly isolated primary hepatocytes is documented to be significantly lower on average with 9.2–68 pmol/mg_protein_·min. Nevertheless, a direct comparison is difficult to achieve, as the detected CYP activities are strongly dependent on the respective substrate and the experimental setup.

Concerning the metabolisation of testosterone, the main CYP3A4 metabolite 6β-OH-T was detected exclusively in CHO-CPR/CYP3A4 C1 with a yield of ~55 nmol after 48 h. Based on the metabolisation kinetics, it is apparent that ~50 nmol 6β-OH-T was formed within the first 24 h. Parental CHO and CHO-CPR C12 cells showed no considerable formation of 6β-OH-T above the background. Despite the demonstrated CYP3A4 specificity, the CHO-CPR/CYP3A4 C1 with 79 ± 2 pmol/mg_protein_·min after 4 h and 26 ± 0 pmol/mg_protein_·min after 24 h rank on the level of cryopreserved primary hepatocytes (10–104 pmol/mg_protein_·min) in relation to applied in vitro hepatocyte systems. This is significantly lower than freshly isolated primary hepatocytes (212–1970 pmol/mg_protein_·min), HepG2-CYP3A4 (600 pmol/mg_protein_·min), or HepaRG (100–1160 pmol/mg_protein_·min) [50]. However, as before, this raises the question of the comparability of different test systems. Further optimisation of the mono-CYP3A4 CHO model, for instance by supplementation or modification with cytochrome b5, could prove useful. Interestingly, we found androstenedione formation in intact cells both with and without measurable CYP3A4 activity. The rate of androstenedione formation and the amount formed after 48 h was increased up to ~2.5-fold in recombinant human CPR-overexpressing CHO compared with parental CHO cells. In the first 4 h, approximately similar amounts of 6β-OH-T and androstenedione around 20 nmol were formed in CHO-CPR/CYP3A4 C1. In contrast to 6β-OH-T, the formation rate of androstenedione did not decrease over time but even increased slightly until the end of the experiment. This resulted in a >3-fold increase in the amount of androstenedione compared with 6β-OH-T after 48 h. Reasons for the time-dependent reduction of 6β-OH-T formation are unknown at this point. A decrease in the activity of recombinant CYP3A4 over time or the consumption of essential reaction equivalents are conceivable causes. Since CHO cells are of ovary origin, background activities in the context of steroid hormone metabolism are not surprising when using steroid hormones as substrate [51,52]. Strikingly, androstenedione is not a main metabolite formed from testosterone by CYP3A4 (only approx. 1/600 compared with 6β-OH-T [53]), and we did not detect androstenedione formation in isolated microsomes from the same cell lines (unpublished data).

We thus conclude that this metabolite is formed via a CYP-independent mechanism different from testosterone. Since recombinant human CPR-overexpressing CHO cells did generate up to a ~2.5-fold increase in androstenedione compared with parental CHO cells, CPR could be involved in an androstenedione formation independent of CYP. The critical role of CPR in steroid hormone metabolism is widely known, as it acts as an electron donor for various CYP enzymes involved in steroid metabolism (e.g., CYP17A1, CYP19A1, CYP21A2) [54]. However, a connection to further CYP-independent hormonal metabolic pathways is less understood. Further evidence for our hypothesis is provided by transcriptome analyses of CHO cells, in which the expression of 22 CYP enzymes was detected [55]. CYP51 and CYP20A1 were the most abundant. CYP51 is involved in cholesterol formation and is, like the above-mentioned CYPs of steroid metabolism, localised in the ER membrane. Almost nothing is known about CYP20A1. Ubiquitously expressed in many tissues, its cell localisation (predicted plasma membrane, cell junctions, actin filaments, and nucleoplasm), substrates, and function have not yet been elucidated. The other detected CYPs are expressed at a significantly lower level. Prominent CYPs of biotransformation (i.e., CYP1A2, CYP2B6, CYP2C9, CYP2C19, CYP2D6, CYP2E1, CYP3A4/5) as well as those of steroid metabolism (CYP17A1, CYP19A1, CYP21A2) were not detected.

The results of this study show that CHO cell systems with combined stable recombinant overexpression of CPR and a specific CYP are well suited for metabolisation studies of drugs and xenobiotics. In the future, they can significantly contribute to elucidating which CYPs are involved in metabolisation and which are not. Furthermore, they can provide information on metabolites produced by a particular CYP and eventually be also used to provide these in smaller quantities as references. This can be performed at the level of intact cells and the microsomal level. Since CHO cells, in contrast to hepatocytes, show only a low expression of cytochrome b5, which has been shown to modulate CYP activities, an additional modification of the stable recombinant expression of cytochrome b5 might be useful in the future [33,37]. This could further increase the performance of the mono-CYP CHO model. However, such mono-CYP CHO cell lines and microsomes require that no interfering basal background CYP activity be detected with regard to other relevant CYPs. In order to exclude this, further activity as well as transcriptome studies could be performed in the course of generating further mono-CYP CHO models in the future.

## 5. Conclusions

In the present study, CHO cell lines with stable and specific CYP activity for CYP1A2, CYP3A4, or CYP2B6 and high proliferation performance in suspension were successfully generated. For certain CYPs, enzyme activity was strongly enhanced by additional recombinant CPR expression. These novel cell lines with specific CYP and/or enhanced CPR activity represent a promising “humanised” in vitro model for background-free studies of the metabolisation activity of a specific CYP enzyme. This could be relevant for various toxicology and or pharmacology studies in the pharmaceutical industry or medicine. This includes the study of phase-1 enzyme reactions as well as drug‒drug interactions when applying cell-free protein expression based on extracts from these CHO cells. Finally, our engineered CHO cells pave the way for future lab-scale production of drug-specific metabolites.

## Figures and Tables

**Figure 1 cells-12-02140-f001:**
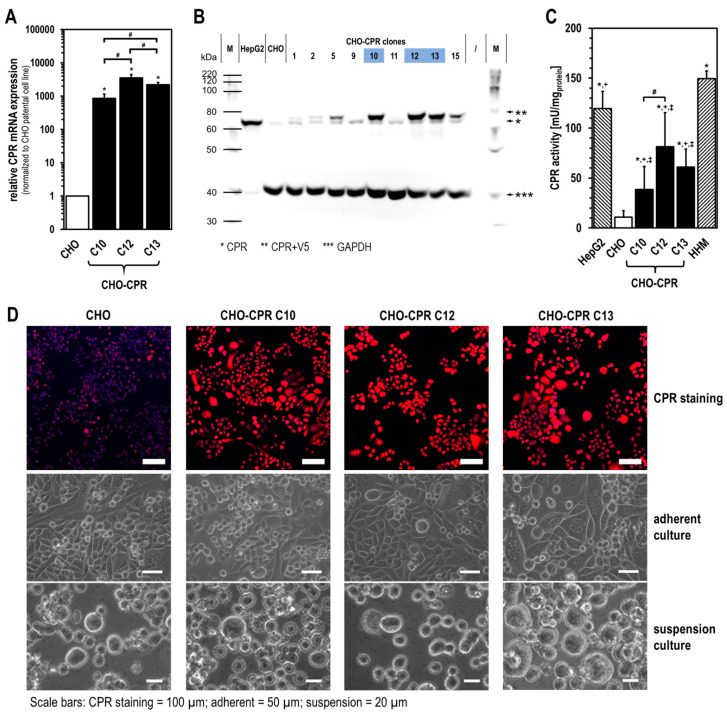
**Recombinant CPR expression and activity in CHO-CPR cells.** CPR expression was analysed at the mRNA level by RT-qPCR (**A**) and at the protein level by Western blotting (**B**). Elevated CPR expression was further validated by indirect immunofluorescence (CPR in red, nuclei in blue) (**D**). CPR activity was determined from isolated microsomal fractions of recombinant CPR-expressing CHO clones compared with HepG2, HHM, and parental CHO cells (**C**) (HHM = human hepatocyte microsomes; mRNA expression with n = 6 and CPR activity data with n = 11 presented as mean ± standard deviation; *p* < 0.05 with * compared with CHO; ^+^ compared with HHM; ^‡^ compared with HepG2; ^#^ compared between CHO-CPR clones).

**Figure 2 cells-12-02140-f002:**
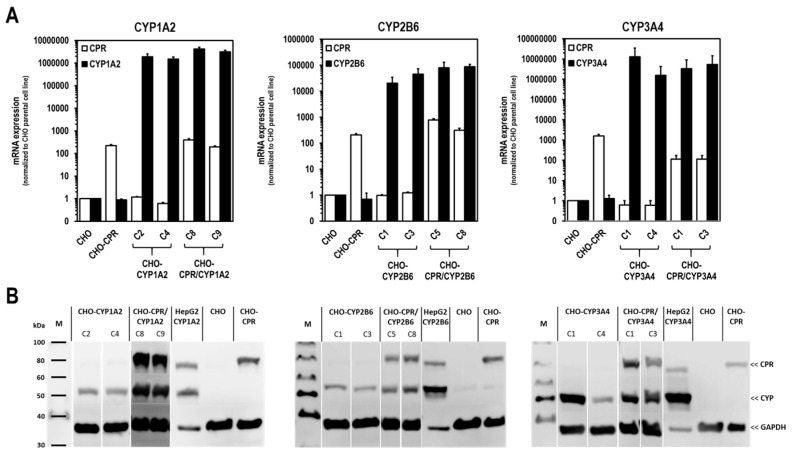
**Recombinant CYP and CPR/CYP mRNA and protein expression in modified CHO cells.** Shown are the selected CHO clones expressing CYP most strongly at mRNA (**A**) and protein level (**B**). In all clones, a specific expression of the respective transfected CYP enzyme was detected. CPR overexpression was also detectable in all CHO-CPR/CYP clones (mRNA expression data presented as mean ± standard deviation; n = 3).

**Figure 3 cells-12-02140-f003:**
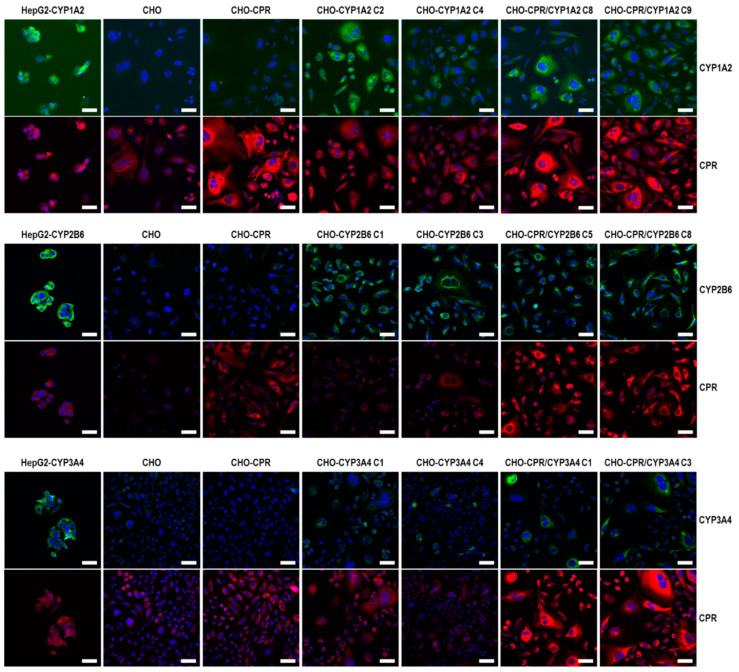
**CPR and CYP protein detection by indirect immunofluorescence.** Specific expression of CYP or CPR/CYP was detected in selected CHO clones. CPR and CYP signals were localised mainly close to the nuclei of the cells (CPR in red, CYP1A2, CYP2B6, and CYP3A4 in green, nuclei in blue, scale bar: 50 µm).

**Figure 4 cells-12-02140-f004:**
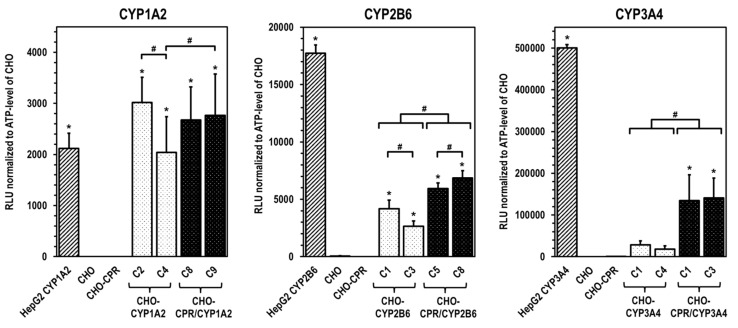
**CYP activity in CHO clones with recombinant expression of human CYP.** Comparative representation of specific CYP activity from CHO-CYP clones with and without recombinant human CPR, compared with the parental CHO and CHO-CPR C12 cell line (data presented as mean ± standard deviation; * *p* < 0.05 compared with CHO; ^#^ compared between CYP clones; n = 12 for CHO based clones; n = 6 for HepG2-CYP overexpressing positive controls).

**Figure 5 cells-12-02140-f005:**
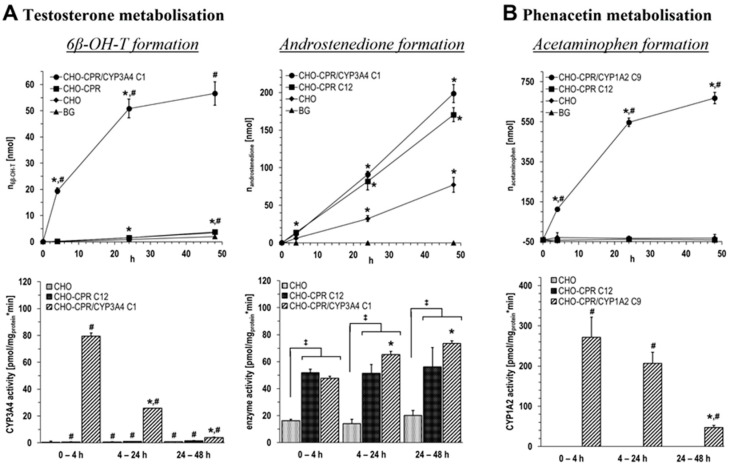
**CHO-CPR/CYP clones specifically metabolised prototypical CYP substrates.** Formation of 6β-OH-T and androstenedione from testosterone by CHO-CPR/CYP3A4 C1 (**A**) and acetaminophen from phenacetin by CHO-CPR/CYP1A2 C9 (**B**) compared with parental CHO and CHO-CPR C12 cells in suspension during 48 h metabolisation (BG: background (substrate without cells); data presented as mean ± standard deviation; two-way ANOVA with Tukey’s multiple comparison test was used to probe for significant differences between groups; *: *p* < 0.05 compared with previous time/section of a sample; ^#^: *p* < 0.05 compared with BG; ^‡^: *p* < 0.05; n = 4).

**Table 1 cells-12-02140-t001:** Primer sequences for gene expression analysis by qRT-PCR.

Gene	mRNA ID	Primer Sequence (5′→3′)	
Reference genes:			
*GAPDH-CHO*	NM_008084.3	fw	GCCAAGAGGGTCATCATCTC
		rev	CCTTCCACAATGCCAAAGTT
*Vezt*	XM_007635120.2	fw	GTGTGAAAGTGGGGCTGAAT
		rev	GTTCCTGCATGGTGGTGAAT
Target genes:			
*CPR*	NM_000941.2	fw	AAGGCGGTGCCCACATCTAC
		rev	TAGCGGCCCTTGGTCATCAG
*CYP1A2*	BC067428.1	fw	CTGGAGACCTTCCGACACTC
		rev	AGGGCTTGTTAATGGCAGTG
*CYP2B6*	BC067430.1	fw	CTCTCCATGACCCACACTAC
		rev	TGTTGGGGGTATTTTGCCCA
*CYP3A4*	NM_017460.6	fw	GTGGGGCTTTTATGATGGTCA
		rev	GCCTCAGATTTCTCACCAACACA

**Table 2 cells-12-02140-t002:** Proliferation capacity of CPR-modified CHO cells in suspension culture.

	Population Doubling Time [h]	Max. Density * (×10^6^ Cells/mL)	Viability * [%]	Growth Conditions/Selection **
CHO	18.3	8.0 ± 1.7	92	none
CHO-CPR 10	25.2	4.6 ± 1.4	89	Blasticidin
CHO-CPR 12	19.9	5.9 ± 0.3	90	Blasticidin
CHO-CPR 13	31.9	3.1 ± 0.1	79	Blasticidin

* after 120 h in suspension culture; ** Blasticidin: 3 µg/mL.

**Table 3 cells-12-02140-t003:** Proliferation capacity of CYP and CPR/CYP modified CHO cells in suspension culture.

	Population Doubling Time [h]	Max. Density * (×10^6^ Cells/mL)	Viability * [%]	Growth Conditions/Selection **
CHO	18.3	8.0 ± 1.7	92	none
CHO-CPR 12	19.9	5.9 ± 0.3	90	Blasticidin
CHO-CYP1A2 C2	22.3	4.4 ± 0.6	97	Zeocin
CHO-CYP2B6 C1	18.4	6.9 ± 0.7	95	Zeocin
CHO-CYP3A4 C1	24.4	3.9 ± 0.2	92	Zeocin
CHO-CPR/CYP1A2 C9	22.5	3.2 ± 0.2	94	Zeocin/Blasticidin
CHO-CPR/CYP2B6 C8	20.1	5.1 ± 0.4	94	Zeocin/Blasticidin
CHO-CPR/CYP3A4 C1	20.9	4.9 ± 0.1	93	Zeocin/Blasticidin

* after 120 h in suspension culture; ** Zeocin: 300 µg/mL; Blasticidin: 3 µg/mL.

## Data Availability

All data are available upon reasonable request from the corresponding author.

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
