# Peer review of "Stable Chinese Hamster Ovary Suspension Cell Lines Harboring Recombinant Human Cytochrome P450 Oxidoreductase and Human Cytochrome P450 Monooxygenases as Platform for In Vitro Biotransformation Studies"

_cells, 2023, doi:10.3390/cells12172140_

Round 1
Reviewer 1 Report
This manuscript presents a study on stable expression of several human P450s (CYP)in Hamster ovary cells together with the human cytochrome P450 reductase (CPR) for in vitro bio-transformation studies.
The paper is well and clearly written. It describes a new system competing a number of preceeding expression systems, some of them more than 30 years old (the system of Doehmer and Seigel in V79 cells).
I have a few remarks and questions:
Page 3 line 141 : not solution but suspension; virus are not soluble in water.
Line 146 : as a non specialist of lentivirus tranvection, one like to learn how long it takes for each step.
Page 10 : how is it that you detect both the CYP and the CPR on the same blot? are you mixing antibodies? The band around 35 : what is it?
You should better describe your western blot detection method. Generally in our lab we used to do separate detection of diverse proteins on blot duplicates.
I hope you will answer this questions?
I am also wondering if you can detect and measure the amount of P450 using the spectral method of Omura and Sato (UV visible differential spectrometry of the P450-CO complex versus P450 FeIII). If it works it would be a nice addition to the experimental data.
The lentivirus expression system has been used before for P450 expression ( for instance : DOI: 10.1016/j.jconrel.2016.08.019 )
As a whole the paper is interesting. Some parts need clarification (at least for people not familiar with these expression methods).
Thus the paper would be acceptable after minor corrections
Author Response
Dear Reviewer 1,
We would like to thank the reviewer for the critical assessment of our manuscript, which helped us a lot in improving it.
We are pleased to send our response to the reviewer in the attachment, in which we address each of the comments of both reviewers individually and present corresponding changes in the manuscript.

Reviewer 2 Report
The manuscript nicely described the feasibility of expressing CYPs enzymes and CYP reductase in CHO cells that could be used for for in vitro biotransformation studies. The available CHO cell line systems can be used for incubations for up to 48 h tested, which overcomes the weakness of short incubation periods of primary hepatocytes. In addition, the new systems do not have endogenous levels of CYP enzymes like HepG cells. A few questions may help enhance the manuscript:
1. The cytochrome C could play a role for CYP catalysis, did authors check the endogenous levels of this protein in CHO cells? This may be discussed also.
2. Was any comparison done between CHO-CYP cells and primary hepatocytes or HepG-CYP cells for production of metabolites?
3. Can Trypan blue staining differentate vital and dead cells?
4. Was the decreased formation of testosterone metabolite and phenacetin metabolite at late times due to depletion of NADPH, CYP degradation, or further metabolism of the metabolites analyzed?
5. Was there any endogenous level of CYP17A1, CYP19A1, CYP21A2 in CHO cells?
Author Response
Dear Reviewer 2,
We would like to thank the reviewer for the critical assessment of our manuscript, which helped us a lot in improving it.
We are pleased to send our response to the reviewer in the attachment, in which we address each of the comments of both reviewers individually and present corresponding changes in the manuscript.

Round 2
Reviewer 1 Report
The experimental details have now been extended making the paper clearer.
Tha authors have answered all the referees questions.
Thus the paper is now acceptable
Reviewer 2 Report
No more comments